# Precarity and Class Consciousness in Contemporary Swedish Working-Class Literature

Magnus Nilsson 

School of Arts and Communication, Faculty of Culture and Society, Malmö University, S-205 06 Malmö, Sweden; magnus.nilsson@mau.se

**Abstract:** This article analyses aesthetical–political strategies for the promotion of class consciousness among workers in a few examples of contemporary Swedish working-class literature from different genres that describe and criticize precarious working conditions. Special attention is given to how these texts engage in dialogue with the notion of the precariat and to the authors' use of decidedly literary forms. One important result is that Swedish working-class writers highlight the heterogeneity among those working under precarious conditions while also arguing that they share certain economic conditions, both amongst each other and with members of other groups (especially the traditional working class). Furthermore, it is argued that the use of literary forms (as opposed to, e.g., reportage or documentary) reflects the absence in the precariat of class consciousness, and the authors' belief that literature can contribute to the creation of such a consciousness.

**Keywords:** working-class literature; Swedish literature; precariat; work; class consciousness

## 1. Introduction

At least since the late nineteenth century, Swedish working-class writers have promoted class consciousness among workers (see Mral 1985; Nilsson 2021a) by contributing to imaginaries, on the basis of which workers make sense of the world and construct class identities (see Eiden-Offe 2017, p. 15; Thompson 1977). Due to continual changes in society's class structure and how these changes have been perceived, these writers have often had to revise their aesthetical-political strategies (Nilsson 2012). For example, after the Second World War, working-class writer Folke Fridell argued that intensified industrialization and the rise of the social-democratic welfare state necessitated the creation of a new kind of working-class literature than that which had its breakthrough in the 1930s and which had mainly depicted the rural proletariat during the early twentieth century (see Nilsson 2014, pp. 45–48). Similar arguments were also advanced in the following decades when commentators argued that Sweden's alleged transformation into a post-industrial and multicultural society had made the critique of class injustice in older working-class literature outdated or even obsolete (see Nilsson 2010). Furthermore, during periods when class and class injustice received little attention in political, cultural or public discourse, interest in working-class literature has often diminished. In the 1950s—when many argued that Sweden, because of the rise of the welfare state, was no longer a class society—and during the 1980s—when neoliberal ideas that were hostile to the very concept of class became near-hegemonic—many even argued that the tradition of Swedish working-class literature had come to an end (see Nilsson 2006, pp. 74–75, 87–88). In the new millennium, however, a "new wave of Swedish working-class literature" (Nilsson 2021b) has emerged, in tandem with an increased interest—especially in literary discussions—in questions about labour and class, which has not least focused on the increasingly precarious working conditions in some sectors and among some groups on the labour market in post-welfare-state Sweden (see Landström 2020, pp. 106–7). One well-known example of this new working-class literature is Kristian Lundberg's novel *Yarden* (2009), which describes the author's

experiences of working under precarious conditions in the harbour in Malmö in Southern Sweden. Lundberg's work has received much attention, both from readers and critics, and in discussions about Swedish class society outside the realm of literature.

The aim of this article is to analyse the aesthetical–political strategies for the promotion of class consciousness among workers developed in contemporary Swedish working-class literature about precarious work. My examples will come from different genres, including novels, graphic novels, reportage and poetry. Special attention will be given to how these works engage in dialogue with the concept of *the precariat*—primarily as defined and promoted by Guy Standing (2011)—and to the authors' use of decidedly *literary* forms.

## 2. Unity in Diversity

According to Standing (2011), the precariat is a class constituted by its members' lack of labour market security. However, he stresses that it is only a class *in* and not (yet) *for* itself. This means that its members do not self-identify as belonging to a class, or, in other words, that they lack class consciousness and, thus, are unable to act politically to defend their interests. One possible reason for this is that the precariat—as described by Standing—is very *heterogenous*, encompassing academics, artists, manual labourers, paperless migrants working in the black economy, people doing "gigs" for platform companies, etc.

This heterogeneity is emphasized in contemporary Swedish working-class literature about precarious work. For example, in her reportage, *Fint hemma* (Pretty at Home), which is based on her experiences of undercover work as a cleaner in private homes, Kerstin Fredenholm (2005, p. 191, my translation) stresses the differences between herself and paperless migrant workers:

> In comparison with someone who is here illegally, without a work permit or residence permit, and who works as a cleaner, I am in a much better position. At least I am entitled to healthcare and can go to a hospital.[1]

This difference is described as being so enormous that it can hardly be grasped— Fredenholm (2005, p. 211, my translation) finds it difficult to imagine "what it is like to be a paperless migrant in Sweden and have to work as a cleaner to survive."[2] Nevertheless, she does try to picture what it would be like:

> What couldn't you be forced to accept in such a precarious situation. Working as a journalist, I have often heard stories about people working for 25–30 kronor per hour, which does not seem to be uncommon at all. Yes, even worse. Ten kronor or no pay at all when they are done with the job. But also, that sexual services are necessary to even get a shitty job.
>
> Who reports such wrongs? Hardly those who are here illegally.
>
> There is also another aspect of this. I have back-up, someone who actually knows where I am and who is aware that there can be a risk with what I do. How often does an illegal immigrant have a back-up that can come to her rescue? Who knows where she is?
>
> (Fredenholm 2005, p. 211, my translation)[3]

In her graphic novel *Wage Slaves* (2016, English translation 2019), Polish-Swedish comic artist Daria Bogdanska relates her experiences of working as a paperless migrant in Malmö. Like Fredenholm, she emphasizes that Swedish workers fare better than migrants, but also that there is a hierarchy among migrants from different countries. After having begun working at a restaurant, she understands that she will have no time for anything but school (she has come to Malmö to study comics) and work, but, immediately thereafter, she realises that her Asian workmate leads an even more stressful life:

> I was suddenly ashamed to be complaining. My school was within walking distance, and it was an artsy-fartsy playground for white Swedes with no home-work or pressure. Nirja was working full-time while studying at university at the

same time. An education paid for by her family in Bangladesh, hoping it would provide her a better life.

(Bogdanska [2016] 2019, p. 35)

Moreover, upon finding out that she will only make DKK 50 per hour, she learns that the people working in the restaurant where she has just paid DKK 15 for falafel probably make even less (Bogdanska [2016] 2019, p. 37). Eventually, the protagonist even formulates a theory about ethnic/national segregation among the workers in the restaurant where she works:

> I started to understand how this place worked. Sanad [the owner] paid the least to those who were the most desperate . . . those who weren't from Europe and couldn't get other work. Immigrants from Europe were also desperate, but we lived closer to home so got paid a little bit more. The Swedes were paid the most even though they also made quite little. They were bribed with free beer and food. Besides, for them the work was usually just extra income on the side.

(Bogdanska [2016] 2019, p. 39)

*Wage Slaves* describes how Bogdanska tries to unionise her co-workers. Her attempts fail, mainly because many of the migrant workers from Asia are—as described in the quote above—too dependent on their employer. When she and a Swedish workmate tell the other workers that they have joined the union, they learn that the others will not join them: "But seriously, Daria, you know it's not gonna work for us. We're not Swedish. It's two different worlds. There's nothing we can do" (Bogdanska [2016] 2019, p. 172). Thus, in *Wage Slaves*, the differences between those working under precarious conditions make it impossible for them to become a class for themselves—a class that can act collectively to defend its interests.[4]

In *Yarden*, Lundberg—much like Fredenholm and Bogdanska—highlights differences between workers and, in particular, differences that relate to migration, ethnicity/religion and legal status. This was accentuated by Åsa Arping (2011, p. 194), who argued that the narrator in *Yarden* situates himself in relation to undocumented migrants, Muslim workers and Philippine sailors. For example, soon after having begun working in the harbour, he realizes that because he is Swedish, he belongs to "the fortunate ones" (Lundberg 2015, p. 72, my translation).[5] Like Bogdanska, Lundberg (2015, p. 99) also describes a hierarchy among the non-Swedish workers, underlining that paperless migrants often have to work for very little money, and that they sometimes do not even get paid at all. In addition, he depicts how the Philippine sailors he meets while working in the harbour lead lives that are so different that the protagonist cannot even begin to understand them (Lundberg 2015, p. 52). Further, unlike Fredenholm, he does not even try to put himself in their position.

Even if Lundberg emphasises differences between those working under precarious conditions, *Yarden* can, nevertheless, be read as an attempt to describe a possible foundation for class consciousness. This is highlighted by literary critic Peter Viktorsson (2009, my translation), who, in his review of *Yarden*, argues that Lundberg gives "voice to a nameless collective, a disparate group of people united through the low-status hard work they do and the low pay they receive."[6]

This writing forth of a collective identity starts with an undermining of the differences between the narrator and the workers from other countries. For example, soon after having declared that the Philippine sailors are so different that they are virtually incomprehensible, the narrator depicts how he enjoys socialising with them (Lundberg 2015, p. 51) and how linguistic and cultural barriers are secondary, since "work understands work" (Lundberg 2015, p. 53, my translation).[7] Further, after having worked in the harbour for six months, he no longer emphasizes what sets him apart from workers from other countries, but instead views himself as "one among all the others. No more and no less" (Lundberg 2015, p. 90, my translation).[8] This is based on them all performing the same work: "The work in itself is so meaningless that it is hard not to become solidaric with each other. We have the same

worth, speak the same language, live with the same fatigue and worries" (Lundberg 2015, pp. 97–98, my translation).[9] Thus, as Arping (2011, p. 194) pointed out, Lundberg shows that beyond all their differences, the workers share at least one thing: their powerlessness on the labour market. The perhaps most interesting expression in *Yarden* of this idea can be found in the following formulation: "we tell on each other in order to secure our own positions" (Lundberg 2015, p. 48, my translation).[10] It can, of course, be read as a highly pessimistic description of a worker collective that is undermined by competition. However, it also highlights—through the use of the pronoun "we"—that precarity is something that is shared by all the workers, and, thereby, it accentuates this as a possible foundation for class consciousness.[11]

A similar picture emerges in Anders Teglund's epistolary novel *Cykelbudet* (2022), which portrays the author's experiences of working as a bicycle courier in Gothenburg during the COVID-19 pandemic. Like the authors mentioned above, Teglund emphasises differences between Swedes and immigrants. For example, when comparing himself to a courier from Afghanistan who is working on a rainy day without any rain clothes, he realizes that he is quite privileged: "I'm reminded of the many privileges I have in my life—health, family, a roof over my head, an economy that allows me to pay my rent. A place in a secure homeland. Rain trousers from Helly Hansen" (Teglund 2021a, p. 192, my translation).[12] But, Teglund also—again, much like Lundberg—highlights what unites Swedish and immigrant workers. For example, when interviewing an immigrant worker, he conveys that he recognises himself in the other's story, despite all the differences between them. Among other things, both of them have felt shame when others have learned that they work as couriers (Teglund 2021a, p. 177). Furthermore, Teglund (2021a, p. 233) stresses that several of the non-Swedish couriers are well-educated and have previously had more high-status jobs, which is also true for himself (before the pandemic he worked as a pianist, as a publisher and as an administrator of a project in the cultural sector). Thus, experiences emanating from *working under precarious conditions* (its low status, the fact that the pay does not match their formal qualifications) unite the couriers, despite the many differences between them.

### 3. Different Kinds of Precarious Work

Teglund not only argues that gig workers, despite the many cultural differences between them, share important experiences of precarious working conditions. He also tries to show that the precarity they experience is also experienced by people usually considered to belong to higher classes. This is explained, primarily, through comparisons between his experiences of cultural work and of working as a bicycle courier. In an article, Teglund (2021b, p. 16, my translation) claims that one of the main things he wanted to achieve with *Cykelbudet* was "to make a comparison between gig work for musicians and couriers, and to explore contrasts and similarities."[13] He also points out that the term gig comes from the world of music (Teglund 2021b, p. 16).

In *Cykelbudet*, Teglund details how incredibly low his revenues from the music streaming service Spotify are, thus underscoring that musicians—just like bicycle couriers—are low-paid workers that are exploited by platform companies (Teglund 2021a, pp. 95, 148, 186, 305). Further, he points out similarities between Spotify's and Foodora's (the platform company for which he works as a courier) business models (Teglund 2021a, pp. 243–45). Teglund often underlines that the insecurity characterising his work as a courier is typical also of his work in the cultural sector. For example, he juxtaposes a description of waiting for the renewal of his contract as a courier with that of his waiting for the result of a grant proposal for a music project (Teglund 2021a, pp. 211, 218). He also points to the fact that it is not only when working as a courier that he is dependent on algorithms. One example of this is his description of a day when he first tries to come up with a way to trick Spotify's algorithm into making his new album more visible, and then discusses with other couriers how one can evade low-paying orders in Foodora's app (Teglund 2021a, pp. 213–16). In addition, he highlights that cultural workers, such as editors of cultural journals or profes-

sional musicians, often make even less money (per hour) than bicycle couriers (Teglund 2021a, pp. 148–49, 186–87, 196). The economic insecurity for many cultural workers is further underlined in his statement that he started working as a courier out of economic necessity when all his gigs as a cultural worker were suddenly cancelled because of the outbreak of the COVID-19 pandemic (Teglund 2021a, p. 31).

Nevertheless, in addition to emphasising the similarities between cultural workers and bicycle couriers, Teglund also acknowledges that there are some rather important differences between the two groups. Not least, he repeatedly underlines that the precarity he has experienced in the cultural sector has not been as severe as that he experienced as a bicycle courier. For example, just after having been hired as a courier, he goes to a meeting with some cultural workers who plan to apply for funding for a project and immediately notices that the offices where they sit are much nicer than those where he applied to become a courier (Teglund 2021a, pp. 44–45). He also highlights that the precarious working conditions he has experienced as a cultural worker are—at least to some extent—a product of his own choices: "I have been given many chances to have some kind of reasonable career, but I think that I have fooled away the possibilities along the way. Now I've probably put myself in an untenable situation" (Teglund 2021a, p. 47, my translation).[14] Thus, a difference between self-chosen insecurity in the cultural sector and the precarity experienced in an "untenable" situation is emphasised.

At the same time, despite highlighting differences both among the gig workers and between different sectors on the labour market, Teglund does not describe any *qualitative* but, rather, a *quantitative* difference between different kinds of precarious working conditions. This idea is summed up in a formulation early in *Cykelbudet*, where Teglund characterises his predicament—being a cultural worker who has to become a bicycle courier—as one of "falling deeper into the precariat" (Teglund 2021a, p. 46, my translation).[15]

## 4. The precariat and the working class

The above descriptions of precarious working conditions as being something that unites people of different backgrounds who work in different sectors constitute a good foundation for the creation of class consciousness. However, such consciousness has not only a positive but also a negative side—oftentimes, classes are defined just as much in terms of opposition to others as in terms of what their members have in common.[16]

According to Standing (2011), the precariat is defined—among other things—in opposition to the traditional working class, which, according to him, enjoys the labour market security that the precariat lacks. Such an opposition is thematized in some of the works analysed in this article. For example, Lundberg highlights how workers with permanent contracts exclude him—who works for a temporary work agency—from the worker collective, and even exploit him: "About me, one can say—'Let him do it! Fuck if we should have to clean up!' and then I do all the things that the others don't even want to think about" (Lundberg 2015, p. 50, my translation).[17] However, many writers also deconstruct the opposition between working class and precariat. This too can be found in *Yarden*, where the narrator connects the hostility shown to him by workers with a permanent contract to the fact that he represents a threat to their labour market security:

> The atmosphere in the workroom is uneasy. I know that I have taken another's place; that I, through what you may very well call my alacrity, have forced away an employee, a person who has suddenly become superfluous. A workmate.
>
> (Lundberg 2015, p. 24, my translation; see also 58 and 115)[18]

Thus, the precariat and the working class are not presented as separate, but as interrelated, groups. The narrator in *Yarden*, furthermore, describes a continuity between the working-class experiences of his youth and his experiences as a middle-aged man of precarious work, which undermines the opposition between precariat and working class: "I am back where it all began for me. Manual labour. Renting out one's muscles" (Lundberg 2015, pp. 57–58, my translation; see also Nilsson 2014, p. 119).[19] The discussion in *Yarden* of

the concept of "class struggle" points in the same direction. The narrator argues that one should speak of two basic groups: "the propertyless" and "[t]hose who own" (Lundberg 2015, p. 58, my translation).[20] In this analysis, there is very little room for differences (or antagonisms) between the working class and the precariat. The narrator in *Yarden* also explicitly connects the existence of the precariat to "the power of capital": "We who show up when it is needed and are then sent back home again; a few hundred-kronor bills richer and a few hours of our life poorer. Never before has the power of capital been so clear to me" (Lundberg 2015, p. 67, my translation).[21] Thus, the narrator describes his and his work mates' relationship to capital as being identical with that of the working class's relation to capital, which, according to Marxist theory, is a relationship between those who own the means of production and those who do not and, therefore, have to sell their labour power.

Another important depicter of precarious working conditions in contemporary Swedish literature—Pelle Sunvisson—also challenges the opposition between precariat and working class by highlighting that the two groups share fundamental conditions and interests. A good example of this can be found in the novel *Svarta bär* (Black Berries, 2021), which describes the realities of Eastern-European workers who have come to Sweden to work as berry pickers (and which is based on the author's experiences of doing the same work under false identity):

> By talking about the berries, the workers could be taken out of the equation in a convenient way. [ . . . ] Berries could not be undignified, could not be exploited, could not contain the risk of injury and ruin. [ . . . ]

> According to Swedish law, the berries were free [ . . . ] and speaking about the price of what is free is kind of pointless. The price of berries was not at all a price for berries, but a price for labour, a price for sometimes injured bodies and occasionally broken dreams, a price for the absence of a parent or a loved one and most of all for the through experience calibrated renumeration at which the pickers were kept in a constant state of hunger.

> (Sunvisson 2021a, p. 61, my translation)[22]

This passage contains a condensed version of Marx's theory of exploitation. Thus, Sunvisson, like Lundberg, highlights that the relationship between migrant workers who work under precarious conditions in contemporary Sweden and capital is the same as the relationship between the working class and capital in Marxist theory. Further, since, for Marxists, it is this relationship that constitutes the working class, this means that the berry pickers are presented as belonging to this class.

There are also examples in many other works about precarious work in contemporary Sweden of how similarities between the precariat and the working class are emphasised. For example, in *Cykelbudet*, a courier tells a colleague not to view those working in restaurants, for example, waiters and cooks, as enemies and claims that "it is important to remember that they are also workers, just like us" (Teglund 2021a, p. 217, my translation).[23] Another good example of how precarious work is associated with the working class can be found in Hanna Petersson's comic "Pigan" (The Maid, 2012). Its protagonist works as a cleaner and describes her employers as "the upper class," while jokingly calling her physically demanding work "a proletarian gym-session" (Petersson 2012, pp. 7, 9, my translation).[24]

## 5. A Return to the Past

According to Standing (2011), the precariat is a new class, constituted by labour market insecurities generated by the flexible capitalism that has developed after the era of welfare state capitalism. Furthermore, he argues that the precariat should be viewed as a class distinct from—and, at least, in part, antagonistic to—the traditional working class. As I have demonstrated above, the latter idea is contested in contemporary Swedish working-class literature, where the distinction between precariat and working class is challenged. Many commentators have also criticised the former claim, that precarious conditions in the labour market are typical for contemporary capitalism. For example, the Marxist sociologist Vivek

Chibber has argued that capitalism has always produced precarity, and that its increase in the Western World in recent decades, thus, represents a return to normality: "A baseline level of insecurity is forced onto workers by capitalism, all the time [ . . . ]. What has happened in the recent past is that institutions that had temporarily acted to decrease that insecurity are being taken apart" (Chibber 2022, p. 38). This idea is also expressed by several contemporary Swedish working-class writers.

For example, the foreword to an edition from 2015 of Lundberg's *Yarden* opens with a discussion on the abolition, in 1945, of the so-called *statare* system (under which farm labourers were hired on one-year contracts and paid mainly in kind). Lundberg emphasises the similarities between this system and the contemporary labour market: "The statare had been vulnerable and insecure, in a way that regular workers were necessarily not. [ . . . ] Today we can see how a similar system has been created as temporary work agencies have gained ground on the labour market" (Lundberg 2015, p. 9, my translation).[25] A few pages later, the comparison continues:

> In practice, a statare was totally dependent on his squire. Insecurity was his daily bread. His employment security was virtually non-existent. The family often moved. Year after year. From one place to another. We at the Yard moved from work to work instead. We always took someone else's place, someone else's work.
>
> (Lundberg 2015, pp. 12–13, my translation)[26]

The same comparison is also made in *Yarden*, for example, when Lundberg uses the term "[d]aglönare" (day labourer) to describe himself and his work mates who work on temporary contracts (Lundberg 2015, p. 55). This term also appears in Anna Arvidsdotter's (2022, p. 40, my translation, emphasis added) poetry collection *Händer att hålla i* (Hands to Hold, 2022), where it is emphasised that it is usually used about historical, rather than contemporary, conditions: "we are, *once again*, day labourers."[27]

Fredenholm too connects contemporary precarious working conditions to the era before the rise of the welfare state, not least through the insertions in *Fint hemma* of excerpts from Ester Blenda Nordström's reportage *En piga bland pigor* (A Maid Among Maids) from 1914, which details the author's experiences of working undercover as a maid on a farm in the early twentieth century. In her foreword to *Fint hemma*, Fredenholm (2005, p. 6, my translation) states that the excerpts from *En piga bland pigor* are intended to make visible similarities between her experiences and those made by Nordström almost a century earlier: "When I read her texts and compare our experiences it feels like a still-standing society, in many respects."[28] She also argues that her use of the old-fashioned word "piga" (maid) can be motivated by the return of old-fashioned conditions: "It is a word that has no place in modern Swedish, but I discovered that reality motivates its existence and fills it with meaning still today" (Fredenholm 2005, p. 6, my translation).[29]

Sunvisson too points to similarities between the contemporary precariat and the working class a century ago. In an interview in a trade-union-membership magazine, he states that today's migrant workers are facing "the same struggle" as the one fought by workers in Sweden a century ago (Josefsson 2021, my translation).[30] Similar thoughts can also be found in *Svarta bär*. There, one of the berry pickers—Alina—discovers an old abandoned mine in the forest, and on a plaque reads about how much ore it has produced. This makes her realise that workers form collectives and that she, too, belongs to one—that of "foreign berry pickers in Sweden":

> Blueberries weren't copper, a few thousand pickers from all over the world weren't two hundred miners that lived together, and the figures did still not say anything about their lives, but it still said something about them belonging together. They said that if their toil could be summed up, if only into the most fraudulent figures, it was collective after all.
>
> (Sunvisson 2021a, p. 96, my translation)[31]

In *Cykelbudet*, Teglund (2021a, p. 109, my translation) also makes comparisons between contemporary precarious work and the working conditions suffered by workers a hundred years ago: "I talk to dad bout working as a courier", he writes, and then continues: "We moan about today's bad working conditions. It is like it was a century ago, isn't it?"[32]

## 6. Precarious Work and Literary Forms

Critical representations of precarious working conditions are relatively common in contemporary Swedish literature, including, or perhaps especially, in works that could be placed in the tradition of working-class literature. One possible explanation for this is quite simply, as Sunvisson argued in an interview, that there is more space in the sphere of literature than, for example, in that of politics to address class political issues, such as the exploitation of migrant workers (Hansson 2022, p. 27; see also Nilsson 2014, pp. 151–53). However, works about contemporary precarious labour also display a (perhaps surprisingly) high degree of *literariness* that requires further discussion.

*Cykelbudet* opens with a facsimile from a thesaurus of the definition of the word "bud" (courier) that features in its title (Teglund 2021a, p. 7). One aim of this seems to be to emphasise that the word has a wide range of possible meanings and that it can give rise to associations of different kinds. For example, it can be associated with subordination, tough living conditions and low status, but also with communication and future prospects. This can be read as reading instruction—as a signal that the author is interested in giving a rich and multifaceted description of his work as a courier—but also that he wants to get an urgent message across to the readers, and that he is not only describing present conditions but also discussing possible future developments. However, it can also be read as a highlighting of the text's status as literature. Nowhere in *Cykelbudet* does it say that the book is a novel (nor that it belongs to any other genre). Thus, it could very well be read as a factual report. However, ambivalence and openness, as well as self-reflexive language use—both features of the text that are brought to the fore through the facsimile from the thesaurus—are generally considered to be more characteristic of literary than, for example, journalistic texts.

In her foreword to *Fint hemma*, Fredenholm (2005, p. 5) emphasises that it is a journalistic work. But she also highlights that she has *staged* the reality that she describes (Fredenholm 2005, p. 5). Furthermore, the book contains several narrative features that are usually associated with fiction rather than non-fiction. One example is—as has been mentioned above—that she uses her imagination and speculates about how other people might feel in certain situations. Another example is that she does not limit herself to reporting about (staged) events in the real world but also describes dreams. For example, she gives a rather detailed summary of a dream about one of her employers:

> I dream that I'm in Mikael Gustavsson's bedroom. He is killing me. I have a rope around my wrists, so I can't move my hands and arms. I'm on my back and Mikael Gustavsson is on his knees, bending over me. His disgusting grin and the bloodshot yellow whites of his eyes are the last things I see as I struggle to breathe. Every time I try to draw a breath of air, I get a burst of semen down my throat. He squirts again and again in my eyes, in my nose, and in my mouth. I struggle for my life. Can't break free. My vision fades. I slowly suffocate.
>
> (Fredenholm 2005, p. 212, my translation)[33]

This might, of course, be a perfectly accurate description of a dream that Fredholm had when working as a cleaner. But it prompts, in the reader, a kind of meaning production that is more literary than one usually expects from a reportage: the text invites the reader to relate the reported facts to the author's violent and grotesque dream. The insertion in *Fint hemma* of uncommented excepts from *En piga bland pigor* could also be viewed as a kind of montage that is more common in literary works than in journalistic texts.

In his foreword to *Svenska palmen*, Sunvisson makes an interesting case for the use of *literary* forms when describing and criticising precarious work and argues that they are

superior to journalistic forms. Thereby, he distances himself from many other writers who, like him, have engaged in precarious work under false identity but have chosen to present their result as reportages, for example, Nellie Bly, Günter Wallraff and the above-mentioned Nordström. His main argument is that while journalism "can open new worlds", fiction "allows us to live the lives of others":

> Put differently, the demand for objectivity makes the journalist the only possible subject of the reportage. For me, fiction's ability to turn the other into a subject has been decisive. My intention throughout my project has been to make so human portraits of the most silent workers on the Swedish labour market that the reader is forced into living their lives.
>
> (Sunvisson 2021b, p. 6, my translation)[34]

Perhaps, however, it is not only literature's power to create subjectivity but also its power to depict what *does not exist* that make literary forms especially suitable for the description and critique of precarious working conditions. Traditional working-class writers were writing for workers that already belonged to a class-conscious class. Thus, even if they did contribute to the construction of class consciousness—by targeting groups of workers that were not yet class conscious, or by promoting new kinds of class consciousness—they could also fulfil a political function through reporting from reality in order to contribute to setting the political agenda for the labour movement and making class injustices known in the public sphere. This could be one reason for the prominence of realist forms in the Swedish tradition of working-class literature—if the task is to communicate facts, then realism is not a bad choice. Authors writing for the precariat are also focusing on real conditions, and they often use realist forms. However, since these conditions are largely new and, thus, perhaps hard to capture with existing literary means, these writers need to be formally innovative. Further, since the precariat is not (yet) class-conscious, they also need to leave the realm of reality/realism and venture forth into fiction and speculation when describing the class consciousness they are promoting.

In *Svarta bär*, Sunvisson not only uses literary forms to describe precarious work, as he also develops these forms. In particular, he challenges the traditional novel form by not focusing on a single protagonist but instead on many individuals.[35] Further, even if these individuals can be said to constitute a collective with shared economic interests, this is not how they view themselves; while describing what they have in common, Sunvisson also highlights the differences between them. Thus, it is hard to view *Svarta bär* as a collective novel of the kind envisioned—but never realized—by older Swedish working-class writers such as Ivar Lo-Johansson.[36] At least, it is not a novel describing any already existing collective but perhaps a novel attempting to *construct*, rather than represent, collectivity.

Already in the first chapter of *Svarta bär*, Sunvisson describes both that there are differences between the berry pickers and that they are aware of them. One example is that Alina, who could be described as the protagonist of the chapter, reflects on the differences between herself and another woman in the group, Oksana (Sunvisson 2021a, p. 8). Toward the end of the novel, however, she discovers that Oksana is a more complex person than she has assumed, that they actually have a lot in common and that they can engage in meaningful discussions (Sunvisson 2021a, p. 112). After having trawled the news media for stories about people from different countries that have come to Sweden to pick berries, Alina comes to the conclusion that, beyond all their differences, they have things in common too, that they "belong together," even if the descriptions of what unites them are full of "enormous holes" (Sunvisson 2021a, p. 96, my translation).[37] Sunvisson tries—as has been demonstrated above—to conjure forth a possible future class identity for these workers, on the basis of their shared economic interests, and such an ambition makes it necessary to venture into the realm of fiction.

Teglund's representation of the precariat also has an orientation toward the future, not least when he argues that precarious conditions are spreading and that more and more

workers *might* become included in the precariat *in the future*. For example, in one article, he puts forward the following argument:

> [T]he working conditions for those delivering food and freelance musicians might very well spread to other sectors. In a few years, perhaps nurses, economists, car mechanics, veterinaries and fire fighters will also be gig workers to higher degree than today.
>
> (Teglund 2021b, pp. 16–17, my translation)[38]

Thus, he uses *speculation*, which is easier to do in literary than in journalistic texts, to establish a possible foundation for class consciousness beyond both the heterogeneity within the precariat and the possible differences between this class and other worker collectives.

**Funding:** This research was founded by the Swedish Research Council grant number 2022-01839.

**Institutional Review Board Statement:** Not applicable.

**Informed Consent Statement:** Not applicable.

**Data Availability Statement:** Not applicable.

**Conflicts of Interest:** The author declares no conflict of interest.

## Notes

1. "I jämförelse med en person som vistas här utan arbets- och uppehållstillstånd och städar har jag ett betydligt bättre utgångsläge. Jag har i alla fall rätt till vård och kan uppsöka ett sjuhus." .

2. "hur det är att vistas i Sverige som papperslös och tvingas städa för att överleva".

3. "Vad kan du inte tvingas gå med på i en så utsatt situation? I mitt arbete som journalist har jag många gånger hört berättelser om människor som jobbar för 25–30 kronor i timmen vilket inte alls verkar vara ovanligt. Ja, ännu värre. Tio kronor eller ingen lön alls när de är klara med jobbet. Men också att det ingår sexuella tjänster för att överhuvudtaget få ett skitjobb. / Vem anmäler sådana missförhållanden? Knappast den som vistas här utan tillstånd. / Det finns också en annan aspekt i detta. Jag har back-up, alltså någon som faktiskt vet var jag befinner mig och känner till att det kan finnas en viss risk i det jag håller på med. Hur vanligt är det att en papperslös städerska har back-up som kan komma till hennes undsättning? Vem vet var hon finns?" This is one of many examples of Fredenholm not just reporting her first-hand experiences, but also using her imagination, thereby pushing the borders between reportage and fiction.

4. It should, however, be noted that when in the quotes above, Bogdanska describes the differences between Swedes and immigrants she often—for example when describing the school she goes to or when discussing the possibilities to join a union– places herself in the former category, and thereby at least complicates the opposition between native and foreign.

5. "de lyckligt lottade".

6. "röst åt ett namnlöst kollektiv, en brokig samling människor med otacksamt slitgöra och låg lön som gemensam nämnare"

7. "arbete förstår arbete".

8. "en bland alla andra. Varken mer eller mindre".

9. "Arbetet i sig är så meningslöst och tungt att det är svårt att inte solidarisera sig med varandra. Vi är lika värda, talar samma språk, lever med samma trötthet och oro.".

10. "vi anger varandra för att säkerställa våra egna platser".

11. A similar idea is expressed in Anna Arvidsdotter's poetry collection *Händer att hålla i* (Hands to hold, 2022), which thematizes experiences of working as a mailman under precarious conditions. "[V]i har inget gemensamt" ([W]e have nothing in common, my translation), it says in one poem, "vi söndrar / sparkar / trampar frenetiskt vatten" (we divide / kick / tread water frenetically), (Arvidsdotter 2022, p. 25, my translation). Here too, the pronoun we emphasises that a collective marked by a lack of solidarity could still be viewed as a collective. The title of the book also describes at least the desire for collectivity and unity.

12. "Jag påminns om hur många privilegier jag har i livet: hälsan, familjen, tak över huvudet, en ekonomi jag kan betala hyran med. En given plats i ett tryggt hemland. Regnbyxor från Helly Hansen.".

13. "en jämförelse mellan gigarbete för musiker och bud, och undersöka kontraster och likheter".

14. "Jag har fått många chanser till något slags rimlig karriär men har nog slarvat bort möjligheterna längs vägen. Nu har jag försatt mig i en ohållbar livssituation.".

15. "falla djupare ner i prekariatet".

16. In Marxist theory, classes are even *constituted* by their opposition to others.

17  "Om mig kan man säga: 'låt honom göra det! Vi ska fan inte behöva städa!' och så gör jag det som alla andra inte vill kännas vid.".

18  "Det är en olustig stämning i arbetslokalen. Jag vet att jag har tagit en annans plats; att jag genom mitt arbete, min beredvillighet om man så vill, har tvingar bort en anställd, en person som plötsligt har blivit överflödig. En arbetskamrat.".

19  "Jag är tillbaka där allt började för mig. Kroppsarbetet. Att hyra ut sina muskler.".

20  "de egendomslösa"; "[d]e som äger".

21  "Vi som kommer när det behövs och sedan skickas hem igen; ett par hundralappar rikare och ett par timmar av vårt liv fattigare. Aldrig har kapitalets makt varit så tydlig för mig.".

22  "Genom att prata om bären lyfte man på ett smidigt sätt ut arbetet och arbetarna ur ekvationen. [ . . . ] Bär kunde inte vara ovärdiga, kunde inte vara exploaterande, kunde inte innehålla risk för skador och ruin [ . . . ] / Bären var genom den svenska lagen gratis [ . . . ] och att prata om priset på det som är gratis är liksom meningslöst. Bärpriset var inte alls ett pris på bär utan ett pris på arbete, ett pris på ibland skadade kroppar och emellanåt brustna drömmar, ett pris på frånvaron av en förälder eller en älskad och mest av allt den genom erfarenheten kalibrerade ersättningen vid vilken plockarna hölls ständigt hungriga."

23  "de också är arbetare på samma vis som oss".

24  "överklassen"; "proletär gympa".

25  "Stataren hade varit utsatt, otrygg, på ett sätt som en vanlig arbetare inte nödvändigtvis var. [ . . . ] Idag kan vi ser hur ett liknande system har skapats i och med att bemanningsföretagen vinner terräng på arbetsmarknaden.".

26  "En statare var i praktiken helt utlämnad åt sin patron. Osäkerheten var hans dagliga bröd. Hans anställningstrygghet var i det närmaste obefintlig. Ofta flyttade familjen. År efter år. Från plats till plats. Vi på Yarden flyttade istället från arbete till arbete. Alltid tog vi någon annans plats, någon annans arbete.".

27  "vi är åter daglönare".

28  "När jag läser hennes texter och jämför våra erfarenheter känns det som ett stillastående samhälle i många avseenden."

29  "Det är ett ord som inte hör hemma i modern svenska men jag upptäckte att verkligheten motiverar ordets existens och fyller det med innehåll än idag.".

30  "samma kamp".

31  "Blåbär var inte koppar, några tusen plockare från hela världen var inte tvåhundra gruvarbetare som levde ihop och siffrorna sa fortfarande inget om deras liv, men de sa något om att de ändå hörde ihop. De sa att om deras möda gick att sumera, om än bara till den allra bedrägligaste av siffror, var den trots allt gemensam.".

32  "Jag pratar med pappa om arbetet som bud"; "Vi ojar oss över att det kan vara så eländigt med arbetsvillkoren idag. Det är ju som för hundra år sedan.".

33  "Jag drömmer att jag befinner mig i Mikael Gustavssons sovrum. Han håller på att döda mig. Jag har rep runt handledederna så att jag inte kan röra händer och armar. Jag ligger på rygg och Mikael Gustavsson står på knä, böjd över mig. Hans vidriga flin och blodsprängda gula ögonvitor är det sista jag ser när jag kämpar för att få luft. Varje gång jag försöker ta ett andetag får jag en skur av sperma rakt ner i halsen. Han sprutar gång på gång i ögonen, i näsan och i munnen på mig. Jag kämpar för mitt liv. Kan inte komma loss. Det svartnar för ögonen. Jag kvävs långsamt.".

34  "kan låta oss leva andras liv"; "göra subjekt av det främmande"; "Annorlunda uttryckt gör kravet på objektivitet journalisten till reportagets enda möjliga subjekt. För mig har litteraturens förmåga att göra subjekt av det främmande varit avgörande. Min avsikt har genom hela arbetet varit att av de allra tystaste arbetarna på svensk arbetsmarknad skapa skildringar så mänskliga att läsaren tvingas leva deras liv".

35  There have been many attempts in Swedish working-class literature to create collective forms. However, most of them have been rather unsuccessful. This is lamented by Ivar Lo-Johansson (1946, p. 188)—the most well-known Swedish working-class writer of all times, who himself did more than most others to promote collective literature—in an essay where he argues that the Swedish working-class novel is fundamentally individualistic.

36  In a programmatic article, Lo-Johansson (1938, p. 752) describes the collective working-class novel as the (yet) "oskriva romanen" (unwritten novel). See also (Landgren 2011).

37  "hörde ihop"; "enorma hål".

38  "De arbetsvillkor som frilansmusiker och matbud har kan mycket väl komma att sprida sig till ytterligare sektorer. Om några år kanske sjuksköterskor, ekonomer, bilrepratörer, vetrinärer och brandmän också i allt högre grad är gigarbetare." In addition to this, he points out that gig workers have common economic interests with other workers, for example concerning legislation about short-term work contracts. (Teglund 2021b, p. 22).

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
