# Peer review of "Precarity and Class Consciousness in Contemporary Swedish Working-Class Literature"

_humanities, doi:10.3390/h12020028_

Round 1

Reviewer 1 Report

I enjoyed this article and thought it was well written and argued. I have noted a few issues below that can be easily fixed:

From the abstract: "Furthermore, it is argued that the use of literary forms is connected to an absence of class consciousness, and to literature’s usefulness when it comes to describing things not yet existing." - I didn't understand this sentence. Maybe "Furthermore, I argue that the use of literary form (as opposed to journalism or reportage) reflects the absence of a single class consciousness; and the utilisation of literary form suggests that literature can make a contribution in generating class awareness." Is that the kind of idea?

The first paragraph on p.2 ends with a few sentences that seem to have been accidentally pasted in from the instructions to authors (starting with "should briefly place the study in a broad context and highlight why it is important... and going to the end of the paragraph). This needs to be deleted!

On p. 6 - the final sentence of the middle paragraph seems to break off rather abruptly: "Thus, the narrator describes his and his work mates’ relationship to capital as being identical with that of the working class, which, according to Marxist theory, is constituted by its exploitation by capital" - is there something missing here? 

Author Response

Thanks for the comments.

1: I've changed the last sentece in the abstract. The new formulation is: "Furthermore, it is argued that the use of literary forms (as opposed to e.g. reportage or documentary) reflects the absence in the precariat of class consciousness, and the authors’ belief that literature can contribute to the creation of such a consciousness."

2: The instructions on page 2 have been deleted

3: On page six I've changes the unclear sentence so that is now reads as follows: "Thus, the narrator describes his and his work mates’ relationship to capital as being identical with that of the working class’s relation to capital, which, according to Marxist theory, is a relationship between those who own the means of productions and those who do not, and therefore have to sell their labour power."

Reviewer 2 Report

This paper contributes significantly to our understanding of an emerging kind of working-class literature in Sweden. It has numerous typographical errors that need attention, however, and it seems to include editorial comments within the text at the top of page 2. In addition, several sentences begin with either "However" or "For example," a stylistic feature that becomes distracting for the reader and is easily fixed. I would also advise giving the Swedish original of translated passages in notes and indicate whether or not a Swedish item has already been translated into English. If it has, quoting that translation (unless it's wrong) would be preferable to offering your own translation. Finally, the discussion on page 10 needs development. How does Sunvisson's challenging the traditional novel's form by not focusing on individuals differ from Ivar Lo-Johansson literary program with the same goal? Is he indebted to Ivar Lo?

Author Response

Thanks for the comments

1: The instructions on page 2 have been deleted.

2: I've given the Swedish quotes to footnotes. (Only one of the works have been translated to English, thus I have to rely on my own translations)

3: The text has been sent to professional language editing (and many "Howevers" have been deleted...)

4: I've added a few comment sabout the relationship between Sunvisson and Lo-Johansson, such as the following: "There have been many attempts in Swedish working-class literature to create collective forms. However, most of them have been rather unsuccessful. This is lamented by Ivar Lo-Johansson (1946, 188) – the most well’known Swedish working-class writer of all times, who himself did more than most others to promote collective literature – in an essay where he argues that Swedish working-class novel is fundamentally individualistic."; "In a programmatic article, Lo-Johansson (1938, 752) describes the collective working-class novel as the (yet) ” oskriva romanen” (unwritten novel). See also Landgren 2011."